# Exploring the Domestication Syndrome Hypothesis in Dogs: Pigmentation Does Not Predict Cortisol Levels

**DOI:** 10.3390/ani13193095

**Published:** 2023-10-04

**Authors:** JoAnna M. Platzer, Lisa M. Gunter, Erica N. Feuerbacher

**Affiliations:** School of Animal Sciences, Virginia Polytechnic Institute and State University (Virginia Tech), Blacksburg, VA 24061, USA; lisagunter@vt.edu (L.M.G.); enf007@vt.edu (E.N.F.)

**Keywords:** shelter dogs, cortisol, stress, pigmentation, morphology, the domestication syndrome hypothesis

## Abstract

**Simple Summary:**

Domestic dogs have a wide variety of colorations, and previous research has found that, in certain breeds, coat color can be linked to behavior. However, it is unknown if coloration is connected to dogs’ stress responses. To explore this question, we studied dogs living under stressful conditions: an animal shelter. We analyzed their urinary levels of cortisol, a stress hormone, to explore whether values from the shelter and on outings with people correlated with their coloration, specifically, their coat color/pattern, nose color, and extent of white spotting. In this preliminary study, we did not find a connection between their cortisol levels and coloration. While more research is needed, these initial findings do not suggest that dogs differ in their stress responses as a result of coloration alone.

**Abstract:**

Previous research has found connections between pigmentation, behavior, and the physiological stress response in both wild and domestic animals; however, to date, no extensive research has been devoted to answering these questions in domestic dogs. Modern dogs are exposed to a variety of stressors; one well-studied stressor is residing in an animal shelter. To explore the possible relationships between dogs’ responses to stress and their pigmentation, we conducted statistical analyses of the cortisol:creatinine ratios of 208 American shelter dogs as a function of their coat color/pattern, eumelanin pigmentation, or white spotting. These dogs had been enrolled in previous welfare studies investigating the effect of interventions during which they left the animal shelter and spent time with humans. In the current investigation, we visually phenotype dogs based on photographs in order to classify their pigmentation and then conduct post hoc analyses to examine whether they differentially experience stress as a function of pigmentation. We found that the dogs did not differ significantly in their urinary cortisol:creatinine ratios based on coat color/pattern, eumelanin pigmentation, or white spotting, either while they were residing in the animal shelter or during the human interaction intervention. These preliminary data suggest that pigmentation alone does not predict the stress responses of shelter dogs; however, due to the small sample size and retrospective nature of the study, more research is needed.

## 1. Introduction

### 1.1. The Domestication Syndrome Hypothesis

Domesticated animals differ considerably from their wild relatives and sometimes within the domesticated phenotypes seemingly unrelated traits correlate with each other, an observation dating back to Charles Darwin [1]. Behaviorally, domesticated animals have reduced reactivity to humans and, as such, are less likely than their wild counterparts to behave fearfully or aggressively towards humans [2]. Physiologically, their stress response systems are also less reactive to the same stressors [3,4]. Morphologically, domesticated animals are often depigmented, with floppier ears, curlier tails, and more neotenic craniofacial structures, among other traits [5,6]. The commonality of these traits and the possibility of a universal causative mechanism underlying this phenomenon is the basis of the domestication syndrome hypothesis [7].

Many research studies support the domestication syndrome hypothesis. Notably, in their investigation of experimental domestication, scientists from the Institute of Cytology and Genetics in Novosibirsk, Russia, found that breeding foxes strictly on the basis of their behavior towards humans, selecting the least fearful and aggressive animals to breed, also led to physiological and morphological changes. The selected line of tame foxes showed increased frequencies of many of the traits included in the domestication syndrome hypothesis; the first morphological change observed was coat depigmentation. This depigmentation occurred in the form of white spotting and brown mottling. Physiological changes occurred as well: foxes that were selected for decreased behavioral reactivity also displayed lower physiological reactivity to human handling. Tame foxes had lower baseline cortisol levels and showed smaller increases in cortisol in response to an acute stressor involving human handling [7]. These seminal results demonstrate that morphological and physiological changes can arise without direct selection for those traits. 

Scientists have explored possible causal mechanisms of the domestication syndrome hypothesis. One plausible explanation is the neural crest hypothesis [8], which posits that animals with reduced behavioral reactivity towards humans likely have a less active hypothalamic–pituitary–adrenal (HPA) axis [3,4]. Consequently, selecting animals for breeding that exhibit reduced behavioral reactivity might produce animals with less responsive HPA axes, and a mildly deficient neural crest might explain this diminished HPA axis responsivity. The neural crest is a group of cells that crucially contribute to the development of various tissues implicated in traits associated with the domestication syndrome, such as pigment-producing melanocytes [8]. A correlation between morphological (e.g., pigmentation) and physiological (e.g., HPA axis reactivity) traits might support the possibility of a causative mechanism underlying the domestication syndrome hypothesis. 

### 1.2. Dog Pigmentation Overview

Many domesticated mammals display a striking variety of colors and patterns, the most pronounced of which is the domestic dog (*Canis lupus familiaris*) [9]. The mutations leading to changes in mammalian pigmentation can occur early in the domestication process, and humans will actively select for novel variations for aesthetic, superstitious, and functional reasons [10,11,12,13,14]. Variation away from wild-type coloration in animals living in their native environments can reduce camouflage and thereby diminish fitness; however, these natural selection pressures are relaxed in the anthropogenic niche [10]. Even modern free-ranging dogs that primarily breed without human intervention, living on the fringes of human society, exhibit variation in pigmentation [15]. 

The wide variety of pigmentation in dogs is due to just two pigments, eumelanin and pheomelanin, and the depigmentation of both. Eumelanin is typically black, but various mutations have resulted in eumelanin being expressed in shades of silvery gray (“blue”), brown (“liver”), or silvery tan (“isabella”) [9]. Pheomelanin is typically reddish yellow, but less-understood *intensity* mutations have resulted in pheomelanin expression in shades of red, orange, tan, yellow, cream, or nearly white [16]. This lightening of eumelanin and pheomelanin is a form of depigmentation, however depigmentation can also result in the total absence of pigment [7]. A very common form of depigmentation is white spotting [17]. White spotting can be thought of as “erasing” pigment: where white spotting is present, no pigment is produced [18]. White spotting causes pink skin and white fur. The expression of white spotting can range from relatively small areas (e.g., a white dot on the chest) to covering nearly the entire dog (e.g., the dog appears almost completely white).

The color of the eumelanin and pheomelanin and the depigmentation of these two pigments are just a few factors impacting the dog’s appearance. Another factor is how these pigments are distributed across the coat [9]; it is this distribution that determines the dog’s coat pattern. Eumelanin and pheomelanin can be distributed in different areas (e.g., alternating stripes of eumelanin and pheomelanin produce the coat pattern known as brindle) and even banded on the same hair together (e.g., agouti). Dogs can also have coats colored entirely by eumelanin or pheomelanin, resulting in, for example, black and yellow Labrador retrievers, respectively. Eumelanin also colors the nose and skin of dogs, such that the shade of a dog’s eumelanin is visible on its nose (except in rare instances when the nose is fully depigmented) even if its coat has no eumelanin.

To summarize, a dog’s coat pattern is determined by the distribution of eumelanin and pheomelanin. If a dog has only one of these pigments present in its fur, that dog will be solid-colored. In cases of phaeomelanistic fur, we can still gather information about the dog’s eumelanin pigmentation from their nose color. Depigmentation further impacts the dog’s appearance: various shades of eumelanin and phaeomelanin are possible and a dog might also display white spotting over any portion of its body. Just as dogs possess a variety of pigmentation phenotypes, they also display many behavioral phenotypes. Recent research suggests there might be a connection between these variables.

### 1.3. Dogs and Stress

Pet dogs experience a multitude of stressors in our anthropogenic world. Entering the animal sheltering system is a well-documented stressor for dogs, likely due in part to the social isolation [19], spatial restrictions [19], and excessive noise [20] they experience in this environment. Cortisol is commonly used to evaluate an animal’s stress levels [21] and its concentration can be measured in several bodily samples, however urine and feces are often preferred as they are considered the least invasive to collect [22,23]. 

Dogs in shelters have higher cortisol levels than dogs in homes [24,25]; as such, those conducting research in animal shelters often focus on identifying interventions that reduce dogs’ stress levels. Some of the most successful interventions involve interactions with humans. Even temporarily removing dogs from a shelter can reduce their stress levels, although spending time in foster homes [26] is a better intervention for decreasing dogs’ cortisol levels than short outings into the community [27]. However, dogs’ cortisol levels vary across individuals during their time in the shelter [28]. Despite evidence that pigmentation can be predictive of dogs’ behavior [29,30,31,32,33,34], no study to our knowledge has investigated whether dogs’ pigmentation correlates with cortisol levels, particularly those living in animal shelters. 

Previous research has found connections between pigmentation and glucocorticoids in other species [35]. As mutations often have pleiotropic effects [36], it is possible that selection for certain pigmentation types has incidentally selected for changes in these animals’ stress response systems. Furthermore, because stress can impact a dog’s behavior [37], it is possible that differential sensitivity to stress can underlie the behavioral differences observed in differently pigmented dogs. Indeed, previous research has found correlations between dogs’ behavior and their pigmentation supporting the plausibility of this hypothesis [29,30,31,32,33,34]. Thus, when we consider the range of morphological and behavioral variability that shelter dogs display, these animals are a useful population to explore questions about morphology, physiology, and behavior in present-day domestic dogs. More specifically, if pigmentation is a predictor of stress susceptibility in dogs, phenotype-based interventions designed to reduce their stress levels can be further explored. In the present study, we utilize the urinary cortisol:creatinine ratios of shelter dogs exposed to human–animal interventions outside of the animal shelter to uncover the relationships between our three pigmentation variables of interest: coat pattern, eumelanin pigmentation, and white spotting.

## 2. Materials and Methods

We visually phenotyped dogs aged six months and older living in American animal shelters that had been enrolled in previous research studies designed to evaluate the effects of short-term outings and weeklong fostering on shelter dog welfare [27,38]. We then utilized the cortisol:creatinine ratio data from these studies in order to analyze the morphological and physiological data to explore our research questions. 

### 2.1. Visual Phenotyping

We classified the dogs according to the visual presentation of their pigmentation using photographic visual phenotyping. Photographs were sourced from animal shelter websites and social media, as well as taken in-person by the research team. Using these photographs, we categorized the dogs according to their coat pattern, eumelanin color, and amount of white spotting. All three variables were independent of one another, such that a dog’s classification in one category had no bearing on its classification in another.

#### 2.1.1. Coat Pattern

We classified the dogs as solid eumelanin (Figure 1A), brindle (Figure 1B), solid pheomelanin (Figure 1C), shaded yellow (Figure 1D), agouti (Figure 1E), black saddle (i.e., saddleback or creeping tan; Figure 1F), and black back (Figure 1G), based on Bannasch et al. [39] and Brancalion et al. [9], according to the apparent distribution of eumelanin and pheomelanin across their coats. Based on the small sample sizes, we pooled together the phenotypes of black saddle (*n* = 8) and black back (*n* = 16) into a single category called “black with tan” and removed the category agouti (*n* = 1) prior to analysis. Our sample included 88 dogs with coats of solid eumelanin, 34 with brindle coats, 42 with coats of solid pheomelanin, 20 with shaded yellow coats, and 24 with black with tan coats (see proportional breakdown in Figure 2). Thus, our coat pattern analysis included 207 dogs. 

#### 2.1.2. Eumelanin Color

We classified the dogs as black (Figure 3A), blue (Figure 3B), liver (Figure 3C), or isabella (Figure 3D) according to the apparent color of eumelanin pigment present on their nose, skin around the eyes and muzzle, and any visible fur expressing eumelanin pigment. Due to the small sample sizes and difficulties in visual discernment between liver and isabella eumelanin (which are genotypically distinct but can overlap phenotypically), we pooled the dogs of liver and isabella eumelanin together into a single category prior to the analysis. Within our sample, 143 dogs had black eumelanin, 34 with blue eumelanin, and 31 dogs had either liver or isabella eumelanin (Figure 2). A total of 208 dogs were included in the eumelanin pigmentation analysis. 

#### 2.1.3. White Spotting

We classified the dogs according to the apparent extent of white spotting on their bodies following the white spotting categorization scheme utilized by Morrill et al. [40] with one modification: the sixth white spotting category was subdivided into two categories due to observations that subjects placed into the single category within the original scheme showed considerable variations. Our two new categories were described as “no or trace white” and “minimal white.” White spotting is a continuous variable; however, for our purposes, the dogs were placed into one of seven categories for analyses. In the order of increasing white spotting, the categories utilized in this study were labeled as no or trace white (Figure 4A), minimal white (Figure 4B), moderate white (Figure 4C), proportional piebald (Figure 4D), scattered color (Figure 4E), high white (Figure 4F), and extreme white (Figure 4G). Because of the relatively small sample sizes and similarities between phenotypes, we pooled together the dogs in the categories of proportional piebald (*n* = 12) and scattered color (*n* = 12), and the categories of high white (*n* = 13) and extreme white (*n* = 4) prior to the analysis. Due to difficulties in phenotyping the dogs with coats of extremely pale pheomelanin or graying fur, we were unable to categorize five dogs in their level of white spotting. Our sample included 40 dogs with no or trace white, 94 with minimal white, 28 with moderate white, 24 with proportional piebald or scattered color, and 17 with high white or extreme white (Figure 2). As such, our white spotting analysis included 203 dogs. 

### 2.2. Cortisol Collection

The dogs in the present study had previously participated in one of two welfare studies about foster caregiving [27,38]: therefore, there were three experimental phases in this study that aligned with those investigations. As such, we classified each dog’s urine sample according to the study in which it was collected (“intervention type”), and, within that study, when the sample was collected relative to the intervention (“intervention phase”). Intervention type was either a brief outing [27] or weeklong fostering [38]. Intervention phase describes the time point from which the cortisol value was derived: pre-intervention (at the shelter), during the intervention (while spending time with humans outside of the shelter), or post-intervention (after the dog was returned to the shelter). We used the dogs’ urinary cortisol:creatinine ratio as it described the dog’s cortisol level without being impacted by its relative hydration.

#### 2.2.1. Brief Outing Intervention

In the study by Gunter et al. [27], the physiological data from 40 dogs at Fulton County Animal Services (FCAS) in Atlanta, GA, 41 dogs from Detroit Animal Care and Control (DACC) in Detroit, MI, and 42 dogs from the Regional Center for Animal Care and Protection (RCACP) in Roanoke, VA, were collected, resulting in 123 subjects participating in the study. During the study, the dogs spent time with a human away from the shelter for approximately 2.5 h in order to investigate the effect of this intervention on the dogs’ welfare. Dogs had varying levels of prior exposure to their caregiver, who was either a community member, shelter volunteer or staff person, or part of the research team. During the three-day study, the dogs experienced their brief outing in the late morning or early afternoon of the second day. Prior to the outing, the dogs’ urine was collected for the first time in the afternoon of the day before and then in the morning prior to the outing. The third collection occurred in the afternoon of the second day, after the dogs had recently returned from their outing. This collection was reflective of the dogs’ experiences during the outing and was our intervention collection time point. Following the outing, the dogs’ urine was collected in the morning and afternoon of the study’s third and final days. These urine samples were then used to measure the dogs’ cortisol:creatinine ratios before, during, and after the intervention. We referred to Gunter et al. [27] for a full description of the intervention and methods used to collect and analyze the cortisol data. 

#### 2.2.2. Weeklong Fostering Intervention

The physiological data from 41 dogs at Charlottesville-Ablemarle SPCA (CASPCA) in Charlottesville, VA, and 44 dogs at the Pima Animal Care Center (PACC) in Tucson, AZ, were collected for the Gunter et al. [38] study resulting in a total of 85 subjects participating. For the study, the dogs lived in foster caregivers’ homes for seven days to study the effects of weeklong fostering on the dogs’ welfare. Foster caregivers were members of the public and shelter volunteers, and, generally, the dogs were unfamiliar with their foster caregivers and others in the household prior to their fostering stay. Dogs’ urine was collected for 17 consecutive days to measure their cortisol:creatinine ratios: five mornings in the shelter prior to fostering, seven mornings in the caregiver’s home, and then five mornings in the shelter after foster care. The urine samples were analyzed using the same methods as described in the brief outing study. 

### 2.3. Analysis

To investigate whether the dogs’ coat pattern, eumelanin pigmentation, or white spotting influenced their cortisol responses, we analyzed the dogs’ cortisol:creatinine ratios from our five study shelters using three linear mixed models in IBM SPSS Statistics (Version 29). In order to utilize these data, despite the differing numbers of collection time points between the studies, the cortisol values were categorized into one of three phases corresponding to the collection time point. Those phases were either prior to the intervention in the shelter (Phase 1), during the intervention (Phase 2), or in the shelter after the intervention (Phase 3). Phase-level analyses of the dogs’ cortisol:creatinine values were previously performed by Gunter et al. [26,27].

Based on this previous research [26,27], each linear mixed model included the following fixed effects as these variables were shown to affect cortisol levels and were entered into the models as covariates: dogs’ weight (kg) and age (months) in addition to the intervention type (brief outing or weeklong fostering) and intervention phase (in-shelter pre-intervention, during the intervention, and in-shelter post-intervention). 

With regard to our present research questions about the dogs’ morphology, one categorical variable was entered into each model as a fixed effect. A five-level categorical variable (i.e., solid eumelanin, brindle, solid pheomelanin, shaded yellow, and black with tan) was employed to describe the dogs’ coat patterns; a three-level categorical variable (i.e., black, blue, and liver or isabella) was employed to describe the dogs’ eumelanin pigmentation; and a five-level categorical variable (i.e., no or trace white, minimal white, moderate white, proportional piebald and scattered color, and high and extreme white) was employed to describe the dogs’ white spotting. 

To disambiguate the known effects of the welfare interventions and changes in cortisol levels observed during the study’s phases from our present research questions, two- and three-way interactions were entered into our linear mixed models as fixed effects. These included an intervention-type-by-phase interaction and an intervention-type-by-phase-by-morphology-variable interaction. While the intervention-type-by-phase interaction was included in our analyses for appropriate model specifications, the results were reported in other publications by Gunter et al. [27,38]. Additionally, dog and intercept were included as random effects and intervention phase was included as a repeated effect. A variance covariance matrix was employed, and a diagonal covariance matrix for the repeated measure of phase. The method of restricted maximum likelihood (REML) was used for estimating variance parameter values, and a statistical significance level of *p* < 0.05 was used throughout our statistical models.

## 3. Results

### 3.1. Descriptive Statistics

In total, 208 dogs from five study shelters (FCAS, DACC, RCACP, CASPCA, and PACC) participated in the study. The dogs had an average weight of 23.6 kg (*SD* = 7.0), average age of 38.2 months (*SD* = 31.2), and average cortisol:creatinine ratio of 20.1 $\frac{\;nmol}{L}\;:\frac{nmol}{L}\times 10^{-6}$ (*SD* = 15.2). Dogs included in our sample were more often female (53.5%).

### 3.2. Linear Mixed Models

The dogs included in the present study contributed 1994 cortisol:creatinine values for the analysis. To utilize the values from both the brief outing and weeklong fostering studies in our linear mixed models, we calculated the average cortisol:creatinine ratios for the intervention phases. Each dog contributed three values: the mean of its samples from the in-shelter period before the intervention, the mean value of sample(s) during the intervention, and the mean of in-shelter values after the intervention. This process yielded a total of 616 average cortisol:creatinine ratio values that were used in the analyses below.

To explore the effects of the coat pattern on the dogs’ cortisol responses, the cortisol:creatinine ratio values were statistically analyzed to detect the possible effects of coat pattern, intervention type (i.e., brief outings or weeklong fostering), intervention phase (i.e., before, during, or after the intervention), and interaction of intervention type and phase, or a three-way interaction of intervention-type-by-phase-by-coat-pattern with the dogs’ age and weight also entered into the model. We found that the dogs’ cortisol:creatinine ratios differed significantly (at *p* ≤ 0.05) as a function of the following variables: intervention type (*p* < 0.001), the interaction of intervention type and phase (*p*-value to be reported in [38]), dog weight (*p* = 0.012), and dog age (*p* = 0.007). However, the dogs’ cortisol:creatinine ratios did not significantly differ as a function of coat pattern (*p* = 0.591), intervention phase (*p* = 0.652) or in an interaction between intervention type, intervention phase, and coat pattern (*p* = 0.295). Table 1 provides the estimated marginal means and standard errors of the dogs’ cortisol:creatinine ratios as a function of coat pattern, intervention type, and intervention phase. Thus, after accounting for the interventions and their phases, we did not detect differences in the dogs’ cortisol levels related to their coat patterns (Figure 5).

To investigate the effects of eumelanin pigmentation on the shelter dogs’ cortisol levels, the dogs’ urinary cortisol:creatinine ratios were analyzed to detect an effect of eumelanin pigmentation, intervention type, intervention phase, an interaction of intervention type and phase, or a three-way interaction of intervention-type-by-phase-by-eumelanin-pigmentation along with the dogs’ age and weight. We found that the dogs’ cortisol:creatinine ratios differed significantly (at *p* ≤ 0.05) as a function of the following variables: intervention type (*p* < 0.001), intervention phase (*p* = 0.02), the interaction between intervention type and intervention phase (*p*-value to be reported in [38]), dog weight (*p* < 0.001), and dog age (*p* < 0.001). However, the dogs’ cortisol:creatinine ratios did not significantly differ as a function of eumelanin pigmentation (*p* = 0.322) or in a three-way interaction between intervention type, intervention phase, and eumelanin pigmentation (*p* = 0.387). The estimated marginal means and standard errors of the dogs’ cortisol:creatinine ratios as a function of eumelanin pigmentation, intervention type, and intervention phase are provided in Table 1. Thus, after accounting for the interventions and their phases, we did not find differences in the dogs’ cortisol responses based on their eumelanin pigmentation (Figure 6). 

To better understand the impact of white spotting on dogs’ cortisol responses, their urinary cortisol:creatinine ratios were analyzed to detect the possible effects of white spotting, intervention type, intervention phase, an interaction of intervention type and phase, or a three-way interaction of intervention-type-by-phase-by-white-spotting along with the variables of dog age and weight. We found that the dogs’ cortisol:creatinine ratios differed significantly (at *p* ≤ 0.05) as a function of the following variables: intervention type (*p* < 0.001), intervention phase (*p* = 0.049), the interaction between intervention type and intervention phase (*p*-value to be reported in [38]), dog weight (*p* = 0.011), and dog age (*p* < 0.001). However, the dogs’ cortisol:creatinine ratios did not significantly differ as a function of white spotting (*p* = 0.830) or in a three-way interaction between intervention type, phase, and white spotting (*p* = 0.234). The estimated marginal means and standard errors in the dogs’ cortisol:creatinine ratios as a function of pigmentation, intervention, and phase included in this model are provided in Table 1. As such, we did not detect an effect of white spotting on the dogs’ cortisol levels either in the shelter or during the human–animal interaction intervention (Figure 7).

## 4. Discussion

This translational study was designed to investigate questions of basic and applied relevance, including the applicability of the domestication syndrome hypothesis to the cortisol responsivity of domestic dogs, allowing for the detection of relationships between morphological variables and physiological stress that might offer insights into improving the welfare of shelter-living dogs. In order to answer these research questions, we utilized data from dogs living in American animal shelters that were exposed to a human–animal intervention during which they left the shelter. These dogs displayed a range of coat patterns, eumelanin pigmentation, and white spotting, and we examined their urinary cortisol:creatinine ratios, as a function of these three pigmentation characteristics, prior to, during, and after the interventions. While this study’s retrospective design and limited sample size placed some limitations on the conclusions we can draw from the results, we found that none of our pigmentation variables predicted the dogs’ urinary cortisol levels at any time during the welfare studies.

The domestication syndrome hypothesis purports that depigmentation in domesticated animals may be linked to an increased resilience to the stressors associated with living in an anthropomorphic niche. Very little research has been conducted about the domestication syndrome hypothesis in domestic dogs, despite their distinction of being the first domesticated species. Nevertheless, our results align with the existing literature. Hansen Wheat et al. [41] demonstrated that three morphological traits associated with the domestication syndrome hypothesis (i.e., white spotting, floppy ears, and curly tails) showed no covariation with expected behavioral traits on a breed level. However, a previous study by this research team [42] revealed that the expected behavioral correlations of domestication, such as reduced fear and aggression as well as increased sociability and playfulness, were less pronounced in modern dog breeds compared to those of ancient breeds. They hypothesized that these findings might be the result of the emphasis breeders have placed on selection for morphological traits in purebred dogs since the Victorian era, overriding the existing correlates of the domestication syndrome hypothesis. 

Our study was designed to complement the existing literature [41,42]. While these investigations focused on breed-level analyses utilizing samples of purebred dogs [41,42], we investigated individual dogs of unknown origins, many of which were likely mixed breed. Furthermore, while studies by Hansen Wheat et al. [41,42] explored the possible correlations between morphology and behavior, our study examined potential relationships between morphology and the physiological stress response. 

It is important to note that, while we lacked information on the likely complex breed heritages of the dogs in our study, previous research suggests that North American mixed-breed dogs often have modern-breed ancestry [40,43]. Therefore, it is likely that the dogs in our study may have been subjected to the same selection pressures prioritizing morphology, leading to a possible and previously proposed decoupling of domestication syndrome hypothesis-associated traits [42]. By utilizing a heterogeneous sample of mixed-breed dogs and exploring the interaction between dogs’ morphological and physiological characteristics, our study offers further insights into the applicability of the domestication syndrome hypothesis to contemporary American dogs living in animal shelters.

Our results, however, do not readily align with the existing body of research examining the interrelationships between pigmentation and behavior in domestic dogs. In contrast to the domestication syndrome hypothesis, several studies (all using modern purebred dogs as subjects) have reported that reduced pigmentation is associated with undesirable behavioral traits. In English cocker spaniels, Korean jindos, and Labrador retrievers, the recessive red mutation has been shown to be associated with increased aggression (spaniels and retrievers; [29,30,31,32]) and fearfulness (jindos; [33]). In Labrador retrievers, the recessive eumelanin-lightening liver mutation was associated with lower trainability [32,34]. Notably, however, in English cocker spaniels, the presence of substantial white spotting aligns with the expectations of the domestication syndrome hypothesis: dogs with more white spotting reportedly exhibit lower levels of aggression compared to their counterparts with less white spotting [30,31].

While our study does not directly assess behavior, it is crucial to acknowledge the influence of the physiological stress response on behavior. A more reactive physiological stress response system can correlate with greater behavioral reactivity (e.g., [44]). Hence, we hypothesized that cortisol levels (or, more precisely, urinary cortisol:creatinine ratios), as an indicator of the stress response, might be higher in dogs with pigmentation phenotypes that prior literature has associated with greater behavioral reactivity. However, in these results, we did not find correlations between pigmentation and cortisol levels. Such findings suggest that the differences in behavior observed in these previous studies might have been driven by some factor(s) other than differences in cortisol production. 

From the perspective of applied welfare, we hypothesized that if pigmentation was related to the cortisol levels of dogs living in shelters (or with changes in the cortisol levels arising in response to a human–animal interaction intervention), this knowledge might be useful for personnel involved in animal rescues and shelters. Considering these organizations’ limited resources, the ability to visually triage dogs upon entry, in order to identify individuals that might be at particular risk for welfare impairments, could be beneficial. Nevertheless, as our results failed to demonstrate that dogs’ stress responses differ as a function of their pigmentation phenotype, shelter staff should not focus on dog pigmentation when assessing dogs’ stress levels in the shelter, and it is unlikely to aid adopters in predicting future behavior when choosing a dog (e.g., [40,45]). 

This study was subject to several limitations that may have influenced our results. Firstly, while cortisol is a widely used physiological indicator of an animal’s response to a stressor(s), it is simply a measure of arousal, regardless of its emotional valence (i.e., physiologically, excitement and anxiety can both be viewed as forms of stress) [46]. Thus, we were unable to differentiate between the eustress and distress the animal experienced, despite our specific interest in distress and its relevance to the domestication syndrome hypothesis. Secondly, this was a retrospective examination using the data previously collected in other studies, which led to some design weaknesses. Despite the significant number of urinary cortisol samples collected from shelter-living dogs that were utilized in our analyses, the number of subjects was likely not large enough to test our statistical models’ interaction terms (which were needed in order to include the known effects of our human–animal interaction intervention). Furthermore, the sample sizes in each category of pigmentation type were unequal, and, in some cases, the total number of dogs within a specific category of a morphological variable, such as coat pattern and white spotting, were limited and required pooling together, which likely reduced our detection abilities. 

The dogs in this study had diverse genetic backgrounds and often unknown life histories, introducing additional variability that may have impacted our cortisol values [47]. Because this investigation was envisaged as a translational study with an applied relevance to dogs living in animal shelters, the human–animal intervention was not standardized, which would have been the preferred approach for more basic explorations assessing the applicability of the domestication syndrome hypothesis with domestic dogs. Lastly, we employed visual phenotyping rather than genotyping to assess pigmentation, acknowledging that different genotypes could produce similar phenotypes [9]. Future studies using closely related dogs with similar life experiences, such as littermates, or dogs from populations that have not undergone recent intense artificial selection for morphological characteristics, such as free-ranging dogs, may be better able to address the possible connection between differential stress responding and morphology within the context of the domestication syndrome hypothesis.

## 5. Conclusions

Utilizing data from a sample of dogs living in animal shelters in the United States, we phenotyped subjects for three pigmentation variables: coat pattern, eumelanin pigmentation, and white spotting. We then examined whether these morphological characteristics predicted the dogs’ urinary cortisol:creatinine ratios while they resided in the animal shelter and in response to a human–animal interaction intervention.

In our investigation, we found that neither coat pattern, eumelanin pigmentation, nor white spotting predicted the dogs’ cortisol levels, suggesting that dogs may not differentially respond to the stressors of the shelter or human interactions as a function of their pigmentation. Thus, these preliminary results do not support assumptions of the domestication syndrome hypothesis related to pigmentation and physiological stress responses in this population of domestic dogs. Nevertheless, future studies should be conducted to determine if these null results were caused by study limitations. 

## Figures and Tables

**Figure 1 animals-13-03095-f001:**
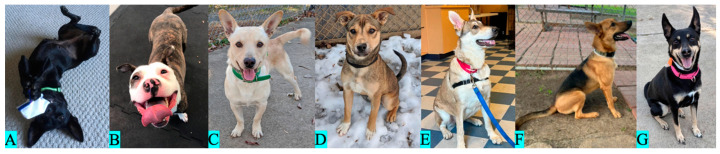
Coat pattern categorization: (**A**): solid eumelanin, (**B**): brindle, (**C**): solid pheomelanin, (**D**): shaded yellow, (**E**): agouti (not included in analysis due to small sample size), (**F**): black saddle, and (**G**): black back.

**Figure 2 animals-13-03095-f002:**
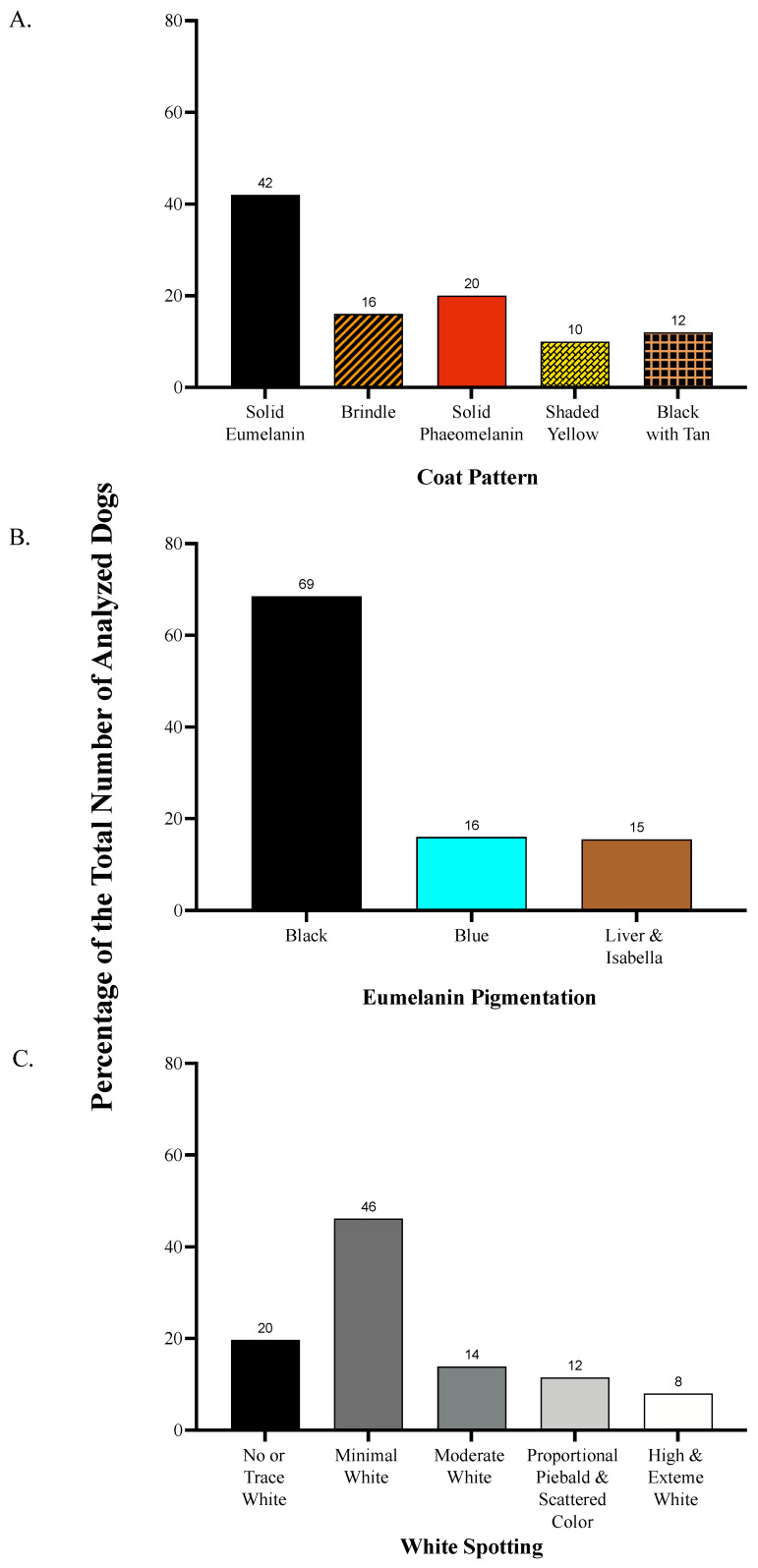
Pigmentation variable distributions in our study samples: (**A**): coat pattern, (**B**): eumelanin pigmentation, (**C**): white spotting.

**Figure 3 animals-13-03095-f003:**
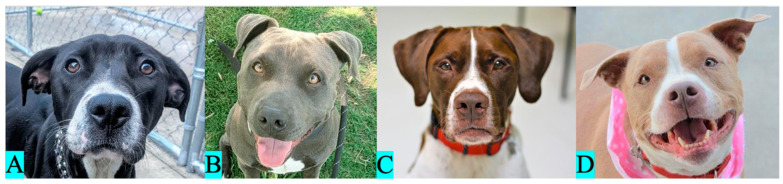
Eumelanin pigmentation categorization: (**A**): black eumelanin, (**B**): blue eumelanin, (**C**): liver eumelanin, and (**D**): isabella eumelanin.

**Figure 4 animals-13-03095-f004:**
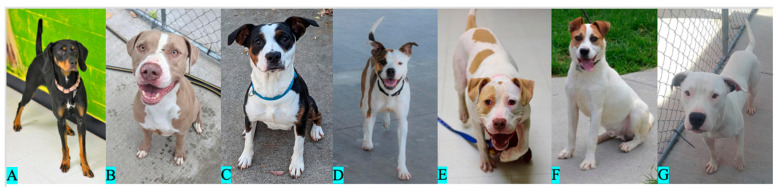
White spotting categorization: (**A**): no or trace white, (**B**): minimal white, (**C**): moderate white, (**D**): proportional piebald, (**E**): scattered color, (**F**): high white, and (**G**): extreme white.

**Figure 5 animals-13-03095-f005:**
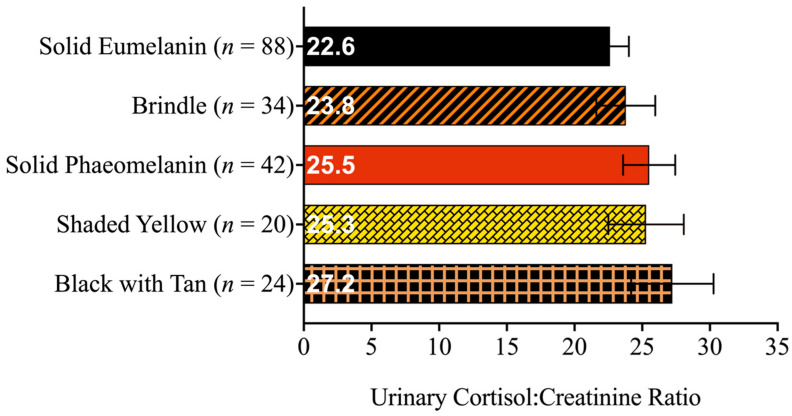
Estimated marginal means of urinary cortisol:creatinine ratios and standard errors of the dogs shown as a function of coat pattern categories. The sample size of each coat pattern category is given next to its label.

**Figure 6 animals-13-03095-f006:**
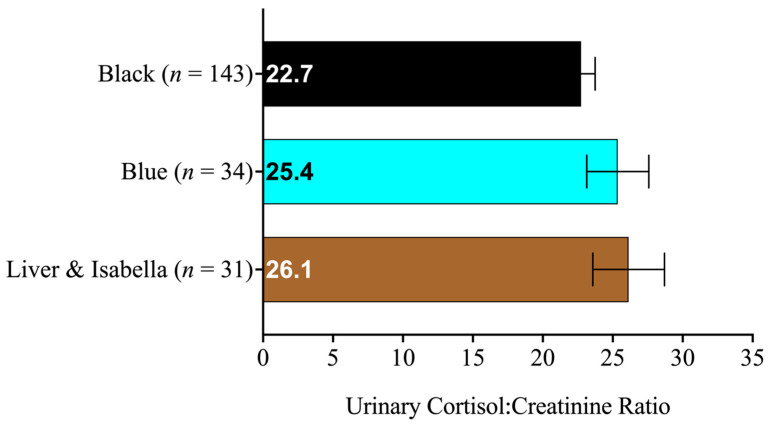
Estimated marginal means of urinary cortisol:creatinine ratios and standard errors of the dogs shown as a function of eumelanin pigmentation categories. The sample size of each category of eumelanin pigmentation is given next to its label.

**Figure 7 animals-13-03095-f007:**
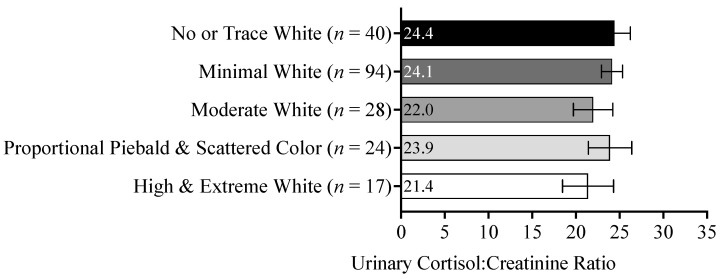
Estimated marginal means of urinary cortisol:creatinine ratios and standard errors of the dogs shown as a function of white spotting categories. The sample size of each category of white spotting is given next to its label.

**Table 1 animals-13-03095-t001:** Estimated marginal means and standard errors of urinary cortisol:creatinine ratios calculated using individual linear mixed models for sub-groups of dogs differing with respect to coat pattern, eumelanin pigmentation, and white spotting.

Pigmentation Variable	Brief Outing Intervention			Weeklong Fostering Intervention		
	Before	During	After	Before	During	After
	*M *(*SE*)	*M *(*SE*)	*M *(*SE*)	*M *(*SE*)	*M *(*SE*)	*M *(*SE*)
Coat Pattern						
Solid Eumelanin	28.3 (1.7)	30.5 (1.8)	27.9 (1.7)	19.0 (2.6)	11.9 (2.7)	18.1 (2.6)
Brindle	29.3 (2.8)	28.9 (3.0)	31.4 (2.8)	19.6 (3.9)	16.3 (4.1)	17.2 (4.0)
Solid Pheomelanin	32.2 (2.5)	36.1 (2.7)	36.2 (2.5)	20.3 (3.4)	11.7 (3.6)	16.7 (3.4)
Shaded Yellow	31.0 (4.8)	39.4 (5.0)	35.3 (4.8)	16.8 (3.8)	13.2 (4.0)	16.0 (3.8)
Black with Tan	38.4 (5.9)	46.7 (6.2)	34.1 (5.9)	18.3 (3.1)	9.7 (3.3)	16.1 (3.1)
Eumelanin Pigmentation						
Black	28.5 (1.5)	33.1 (1.6)	29.6 (1.5)	17.8 (1.6)	11.3 (1.7)	16.2 (1.6)
Blue	30.1 (2.7)	29.4 (2.9)	31.5 (2.8)	22.9 (4.0)	16.0 (4.3)	22.2 (4.1)
Liver or Isabella	34.6 (2.7)	34.6 (2.8)	35.7 (2.7)	22.2 (5.0)	13.5 (5.2)	16.3 (5.0)
White Spotting						
No or Trace White	35.3 (2.9)	39.2 (3.0)	35.0 (2.9)	14.0 (2.9)	10.0 (3.1)	13.0 (3.0)
Minimal	29.5 (1.7)	31.2 (1.8)	32.2 (1.7)	19.7 (2.1)	13.2 (2.2)	19.0 (2.1)
Moderate	26.7 (3.2)	31.2 (3.3)	26.2 (3.1)	19.4 (3.9)	12.6 (4.2)	15.7 (4.0)
Proportional Piebald or Scattered Color	27.5 (3.2)	30.0 (3.4)	25.9 (3.2)	27.6 (4.6)	12.2 (4.8)	20.2 (4.6)
High or Extreme White	25.9 (3.9)	27.7 (4.1)	28.5 (3.9)	17.6 (5.3)	12.2 (5.6)	16.5 (5.3)

Estimated marginal means (Ms) and standard errors (SEs) of values of urinary cortisol:creatinine ratios obtained for dogs classified into various pigmentation categories as a function of intervention type and phase.

## Data Availability

Publicly available datasets were analyzed in this study. These data can be found at: https://data.lib.vt.edu/articles/dataset/Morphology_Cortisol_Data_for_Repository/24101649/1 (accessed on 14 September 2023).

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
