# Peer review of "Exploring the Domestication Syndrome Hypothesis in Dogs: Pigmentation Does Not Predict Cortisol Levels"

_animals, 2023, doi:10.3390/ani13193095_

Round 1

Reviewer 1 Report

This study by accomplished researchers in the field is an attempt to relate changes in pigmentation and appearance of dogs housed in shelters to their cortisol levels. This is an interesting and innovative idea. The study utilizes data from past published work. There is a careful assessment and categorization of pigment patterns, the presentation is generally quite clear, and other aspects of the methodology are strong.   

My major concern regards the findings and ability of the study to detect differences. The authors find no association between pigmentation (coat patterns, eumelanin color, white spotting) and cortisol levels. Studies reporting a lack of significant differences certainly can be valuable, but how much so depends on how likely the study would have detected differences if they did, in fact, exist. In light of the relatively small final samples sizes in most groups, especially given the numerous unrelated environmental and experiential factors that can affect the response to shelter housing, the results to me seem very preliminary. It is an interesting first step, but the preliminary nature of the findings needs to emphasized and the sample sizes need to be discussed as a major limitation.

I also have concerns with how the study is introduced and justified. The Abstract indicates that the main goal of the study is a test of the domestication syndrome hypothesis, specifically “to explore if there is a connection between dog’s response to stress and their coat color/pattern…”. I see the study poorly designed for that purpose. If that were the main goal, one would want to choose a very standard stressor, not an event such as housing in an animal shelter, which is a widely variable combination of various stressors that can vary from shelter to shelter and time to time. Later we learn there also is an applied goal related to assessing welfare of shelter dogs. The design of the study seems to directly address the applied goal and only have implications for the basic goal.

In addition, the rationale presented for the relevance of the domestication syndrome hypothesis is confusing. In the second paragraph of the Introduction, a change in coat color is said to be an early characteristic of domestication (the fox study). If so, why would we expect coat color to account for variation in the traits of a species that has been domesticated for thousands of years? This needs to be spelled out clearly early in the Introduction.

Phenotyping was done visually rather than genetically. That seems reasonable to me for addressing the applied goal of the study. However, some measure of consistency of the process is lacking. All dogs were phenotyped from photos, but 78% were also done in-person. What was the agreement for this 78%.

The conclusion on lines 454-457 should be tempered in light of concerns about samples sizes.

Minor points:

The statement in the Abstract that “adopters should appreciate characteristics besides morphology….” struck me as gratuitous advice that doesn’t belong in the paper. This should be reworded.  

Line 67: “weakened” neural crest is imprecise. Do the authors mean not fully differentiated?

Lines 277-281. The fact that 208 dogs were included in the study is said twice.

Lines 287-288. If one of 208 dogs were eliminated from coat pattern analysis, 207 should be left.

Reviewer 2 Report

This is an interesting paper investigating the possible interrelationships between various aspects of body pigmentation and stress responses of American shelter dogs. It reports negative results (absence of correlations between various features of pigmentation of shelter dogs and their stress responses evaluated by the measurements of urinary cortisol:creatinine ratios). Nevertheless, these results broaden our knowledge concerning the usefulness of focusing on pigmentation during preliminary evaluation of welfare risks for dogs newly admitted to a shelter. These findings also throw important light on the applicability of the Domestication Syndrome Hypothesis in the research devoted to physiology and behaviour of the domestic dog.

However, the text needs extensive editing, as very numerous statements are so much unprecise that this unprecision amounts to serious factual errors. To give just one example, see the following statement: "We found the variables of intervention type (p < .001), weight (p = .012), and age (p = .007) were statistically significant, while the variable of intervention phase was not (p = .652)". Variable in itself cannot be significant or non-significant!

I also heartily recommend to modify the Figure 4, as in its present form it is very difficult to read.
I also provided detailed comments (in total 203) in the pop-up notes that can be found directly on the pdf of the manuscript.

Attention: that PDF should be opened by means of the programme Adobe Acrobat. If it is opened in the programme Edge, the comments in the pop-up windows may have formatting problems.

Good luck with the revision!

English language badly needs to be improved, as very numerous statements are not sufficiently precise and these linguistic imperfections are in many cases equivalent to grave factual errors. I reported many such cases (usually together with the suggestions of necessary linguistic modifications) within the pop-up notes put directly on the reviewed manuscript.

Round 2

Reviewer 1 Report

The authors have been responsive to my concerns.